# Head and Neck Cancer Types and Risks of Cervical–Cranial Vascular Complications within 5 Years after Radiation Therapy

**DOI:** 10.3390/jpm12071060

**Published:** 2022-06-29

**Authors:** Chi-Hung Liu, Bing-Shen Huang, Chien-Yu Lin, Chih-Hua Yeh, Tsong-Hai Lee, Hsiu-Chuan Wu, Chien-Hung Chang, Ting-Yu Chang, Kuo-Lun Huang, Jian-Lin Jiang, Joseph Tung-Chieh Chang, Yeu-Jhy Chang

**Affiliations:** 1Stroke Center, Department of Neurology, Chang Gung Memorial Hospital, Linkou Medical Center, Taoyuan 33333, Taiwan; ivanliu001@cgmh.org.tw (C.-H.L.); thlee@cgmh.org.tw (T.-H.L.); serenawu@adm.cgmh.org.tw (H.-C.W.); cva9514@gmail.com (C.-H.C.); littlefur@cgmh.org.tw (T.-Y.C.); drkuolun@adm.cgmh.org.tw (K.-L.H.); cancer1022@gmail.com (J.-L.J.); 2Department of Medicine, College of Medicine, Chang Gung University, Taoyuan 33333, Taiwan; beanson.tw@gmail.com (B.-S.H.); qqvirus1022@gmail.com (C.-Y.L.); foliatus@gmail.com (C.-H.Y.); 3Department of Radiation Oncology, Proton and Radiation Therapy Center, Chang Gung Medical Foundation, Linkou Chang Gung Memorial Hospital, Taoyuan 33333, Taiwan; 4Taipei Chang Gung Head & Neck Oncology Group, Chang Gung Memorial Hospital Linkou Medical Center, Taoyuan 33333, Taiwan; 5Particle Physics and Beam Delivery Core Laboratory of Institute for Radiological Research, Chang Gung Memorial Hospital, Chang Gung University, Linkou Medical Center, Taoyuan 33333, Taiwan; 6Department of Neuroradiology, Chang Gung Memorial Hospital, Linkou Medical Center, Taoyuan 33333, Taiwan; 7Chang Gung Medical Education Research Centre, Taoyuan 33333, Taiwan

**Keywords:** carotid artery stenosis, head and neck cancer, nasopharyngeal carcinoma, re-irradiation, radiation therapy, proton therapy

## Abstract

Background and purpose: to investigate the frequency of cervical–cranial vascular complications soon after radiation therapy (RT) and identify differences among patients with various types of head and neck cancer (HNC). Methods: We enrolled 496 patients with HNC who had received their final RT dose in our hospital. These patients underwent carotid duplex ultrasound (CDU) for monitoring significant carotid artery stenosis (CAS). Brain imaging were reviewed to detect vertebral, intracranial artery stenosis, or preexisted CAS before RT. Primary outcome was significant CAS at the internal or common carotid artery within first 5 years after RT. We categorized the patients into nasopharyngeal carcinoma (NPC) and non-NPC groups and compared the cumulative occurrence of significant CAS between the groups using Kaplan–Meier and Cox-regression analyses. Results: Compared to the NPC group, the non-NPC group had a higher frequency of significant CAS (12.7% vs. 2.0%) and were more commonly associated with significant CAS after adjusting the covariates (Adjusted hazard ratio: 0.17, 95% confident interval: 0.05–0.57) during the follow-up period. All the non-NPC subtypes (oral cancer/oropharyngeal, hypopharyngeal, and laryngeal cancers) were associated with higher risks of significant CAS than the NPC group (*p* < 0.001 respectively). Conclusion: Significant CAS was more frequently noted within 5 years of RT among the patients with non-NPC HNC than among the patients with NPC. Scheduled carotid artery surveillance and vascular risk monitoring should be commenced earlier for patients with non-NPC HNC. By contrast, vascular surveillance could be deferred to 5 years after RT completion in NPC patients.

## 1. Introduction

Head and neck cancer (HNC) is a group of cancers developing in the soft tissues, salivary glands, and/or the mucosa of the upper respiratory or digestive system that cover the oral and nasal cavities, laryngeal, pharyngeal, hypopharyngeal, and paranasal sinuses’ mucosa. Radiotherapy (RT) is one of the standard treatments as primary or adjuvant setting for HNCs. Late effects on “bystander” organs have become increasingly prevalent among survivors of HNC. Radiation vasculopathy with accelerated atherosclerosis and increased risk of carotid artery stenosis (CAS) have been reported as long-term consequences of radiation injury [1,2,3]. Patients with HNC who present with such conditions after RT are at an increased risk of ischemic stroke [4,5]. RT-induced CAS is typically widespread [6], rapidly progressive [7], and usually affects the common carotid artery (CCA) [8], whereas intracranial vessels are less commonly affected in these patients [8]. A study suggested that carotid artery screening should be conducted yearly starting 5 years after RT [9]. Whether certain groups are at higher risk of developing significant CAS within 5 years of RT has not been thoroughly investigated.

RT techniques have been revolutionized in the past decade, and advancements have been made in photon-beam therapy, such as intensity-modulated RT (IMRT), including volume-modulated arc therapy (VMAT) [10], the most advanced form of IMRT for the treatment of HNC. However, IMRT or VMAT is inherently limited by the physical properties of the photon beam, which results in unavoidable irradiation of normal tissues at low to moderate doses even at considerable distances from the tumor [11,12,13]. Proton-beam therapy (PBT) is an increasingly popular form of RT for the treatment of patients with HNC [14,15]. Protons have physical characteristics of Bragg Peak that deposit most of their radiation dose in a highly confined area. The integral dose with PBT is approximately 60% lower than that of any photon-beam technique. Theoretically, PBT may reduce the integral radiation dose to normal tissues, thereby avoiding collateral damage. However, the incidence of cervical–cranial vascular complications in the patients undergoing these advanced RT methods remain unknown. In the present study, we investigated the frequency of cervical–cranial vascular complications among patients with various HNC types in current era and hoped to develop a precise clinical follow-up strategy of vascular complication between different HNC types.

## 2. Methods

### 2.1. Patient Recruitment and Demographic Data

Our cohort included patients with HNC who had ever undergone RT between 1 November 2011, and 31 October 2021, and who received regular follow-ups at the radio-oncology and neurology departments of Chang Gung Memorial Hospital at Linkou. Because we aimed to investigate neurovascular complications during the early phase after RT and hoped to minimize the bias caused by the variance of RT methods in different era. We retrospectively reviewed the patients completed their RT after 1 January 2015 in this study, who also had the time interval between the final date of RT and the latest date of follow-up ≤5years. Patients who had missing data or had never received carotid duplex ultrasound (CDU) after RT were further excluded. We also reviewed the patients’ pretreatment magnetic resonance imaging (MRI) data to exclude patients who had presented with significant CAS or vertebral artery (VA) stenosis before RT. We recorded the demographic data and comorbidities (dyslipidemia, hypertension, diabetes mellitus [14], cigarette smoking, betel quid chewing, history of coronary artery disease, chronic kidney disease, atrial fibrillation, and ischemic stroke) of all the recruited patients. Dyslipidemia was defined when patients’ low-density lipoprotein cholesterol (LDL) level was ≥130 mg/dL or when they had received lipid-lowering therapy at first visit in to the neurology department. We defined the patients as having histories of cigarette smoking and betel quid chewing if they were current or former users. The patients’ laboratory data, such as glycated hemoglobin, high-density lipoprotein cholesterol, LDL, and free T4 levels were recorded (Figure 1). The study was approved by the Ethics Institutional Review Board of Chang Gung Memorial Hospital (202101981B0 and 202200464B0).

### 2.2. Cancer Treatment and RT Data

The standard treatment in our hospital was the definitive RT for the NPC patients, while around four in five of the larynx, hypopharynx, and oropharynx cancer patients received definitive RT for initial organ preservation as well. Differently, most of the oral cavity cancer patients received postoperative RT in our hospital, and definitive RT was the alternative treatment for patients who were not suitable for surgical treatment. For all the HNC patients receiving RT, platinum-based (Cisplatin or carboplatin) concurrent chemotherapy was the main regimen. The pathological types, locations, and tumor stages of HNC, method of RT (VMAT or PBT), interval from latest RT to study enrollment, and the accumulated total doses of RT were recorded for each patient. 

In VMAT, the planned target volume must account for radiation administered to a margin of at least 3–5 mm around the clinical target. Treatment consisted of 6000–6996 centi-grays (cGy) in 30–33 fractions over 6–7 weeks and 3 days and was determined according to operation or not; five fractions were delivered per week. All the targets were treated simultaneously [16]. For the patients receiving PBT, the treatment plans were generated using the Eclipse planning system (version 13.7; Varian Medical Systems, Palo Alto, CA, USA) with the pencil beam line scanning system. We used three beam angles for full-field PBT plans [10].

### 2.3. Grouping

The patients were categorized into the NPC and non-NPC groups according to cancer pathology. The NPC group consisted of the patients with NPC, whereas the non-NPC group consisted of the patients with oral cavity, oropharyngeal, laryngeal, and hypopharyngeal cancers.

### 2.4. Outcomes and Follow-Ups

The primary outcome in this study was the diagnosis of significant CAS within first 5 years after RT. We defined significant CAS as >50% stenosis on B-mode images with a peak systolic velocity ≥ 120 cm/s based on the hemodynamic criteria for any internal or common carotid artery in the CDU examination. The degree of CAS was determined according to the standard ultrasound criteria [17]. Personnel from our CDU laboratory diagnosed CAS with an overall accuracy of >90% [18]. Routine CDU evaluation prior to the initiation of RT was not recommended in previous guideline [19], we reviewed the images of carotid and vertebral arteries (VA) in the contrast-enhanced CT and/or magnetic resonance imaging (MRI) performed before the initiation of cancer treatment to detect preexisted significant CAS or VA stenosis. We defined as a pre-existed significant CAS/VA stenosis if there were any plaques/stenosis occupied more than 50% of the vessel lumen in the axial views. We hoped this could help to verify whether the significant CAS developed after RT or existed prior to RT.

The secondary outcomes were the presence of significant vertebral artery (VA) stenosis, significant intracranial arterial stenosis (ICAS), and occurrence of symptomatic ischemic stroke (IS). Types of carotid interventions were also recorded.

The patients received regular follow-up at the radio-oncology department at least every 6 months after RT. Contrast-enhanced CT and/or MRI were arranged in the meantime to identify potential distal metastasis or recurrence of the primary tumor. To monitor the neurovascular complications, patients were also referred to our neurology out-patient department during the post-RT period. Upon the first visit at neurology department, the patients’ laboratory data including complete blood count, blood chemistry tests, lipid profile, and thyroid hormone levels were checked. Meanwhile, patients underwent the first carotid duplex ultrasound (CDU) study to screen the presence of significant CAS. Total plaque scores (TPS) [6], intimal medial thickness (IMT) of the common carotid arteries, and degree of CAS were also measured in this examination. Patients received repeated CDU exam 12–24 months later based on the severity of carotid artery lesions in the first visit [19]. We also reviewed the patients’ CT/MR images arranged for cancer follow-ups to find the possible clues of secondary outcomes. 

In patients noted with a sign of significant CAS on any CDU study, significant VA stenosis or ICAS, we then arranged CT angiography or MR angiography to confirm the diagnosis. 

### 2.5. Statistical Analysis

We used SPSS 22.0 (SPSS, Chicago, IL, USA) to analyze the clinical data. Parameters were presented as means ± standard deviations or frequencies (%). We used an independent two-sample *t* test to identify differences in the continuous variables between the study groups. The categorical variables were compared using a chi-square test or Fisher’s exact test. Event risk and time to significant CAS were compared between the study groups through Kaplan–Meier analysis with log-rank test. According to previous reports, we selected known predictors factors for atherosclerosis-associated CAS (smoking, diabetes mellitus, dyslipidemia, hypertension, coronary artery disease, and glycated hemoglobin) and RT-associated CAS (RT doses, types of RT, and re-irradiation) in the Cox regression model [6,20]. We analyzed continuous variables (RT dose and glycated hemoglobin) as continuous data instead of categorized them into groups. We used univariate Cox regression model and multivariable Cox regression model with backward selection to see the relationship between these 9 risk factors, NPC, and CAS risk. Significance was indicated by *p* < 0.05.

## 3. Results

Our cohort included 860 patients with HNC who had ever received RT between 1 November 2011, and 31 October 2021, and had undergone regular post-RT follow-up in our hospital. From this cohort, 554 patients who completed their RT after 1 January 2015, and had the time interval between the final date of RT and the latest date of follow-up ≤5 years were retrospectively reviewed. A total of 54 patients who had never received carotid duplex ultrasound study and 4 patients with missing information were also excluded. Finally, 496 HNC patients were recruited. Among the 496 patients, 201 (40.5%) and 295 (59.5%) were categorized into the NPC and non-NPC groups, respectively (Figure 1). Compared with the patients in the NPC group, those in the non-NPC group were older (non-NPC vs. NPC: 58.42 ± 9.74 vs. 50.01 ± 10.84 years, *p* < 0.01), male predominant (91.2% vs. 82.5%, *p* < 0.01), and had higher frequencies of smoking (56.8% vs. 38.3%, *p* < 0.01) and betel quid chewing (42.8% vs. 14.0%, *p* < 0.01). The non-NPC group also had lower frequency of PBT (15.9% vs. 42.3%, *p* < 0.01) and lower mean RT doses (6624.07 ± 688.90 vs. 6808.82 ± 997.26 cGy, *p* = 0.02).

Compared with the NPC group, the non-NPC group had a lower mean level of low-density lipoprotein (non-NPC vs. NPC: 109.84 ± 44.66 vs. 125.54 ± 44.50 mg/dL, *p* < 0.01), and high-density lipoprotein (49.55 ± 15.39 vs. 54.81 ± 16.08 mg/dL, *p* < 0.01). The mean time interval between RT completion and the first CDU were similar between the two groups (21.79 ± 14.14 vs. 22.82 ± 14.35 months, *p* = 0.44). Moreover, the mean IMT in the CDU study at enrollment was thicker in right side in the non-NPC group (0.76 ± 0.36 vs. 0.68 ± 0.24, *p* < 0.01). The other metabolic parameters, namely the glycosylated hemoglobin and free T4 levels (Table 1), were similar between these study group. 

The mean interval from the RT completion to the last follow up were similar between the two groups. None of the patients noted with significant CAS during follow-up had pre-existed lesion prior to RT. Compared with the non-NPC group, the NPC group had a lower frequency of significant CAS (NPC vs. non-NPC: 2.0% vs. 12.7%; Table 2) and were at a lower risk of significant CAS diagnosis during the follow-up period in the Kaplan–Meier analysis (log-rank test *p* < 0.001; Figure 2), and the Cox regression model (adjusted hazard ratio [AHR] = 0.17, 95% confident interval [CI] 0.05–0.57, *p* = 0.004). Of note, the glycated hemoglobin level (AHR = 1.03, 95% CI 1.01–1.06, *p* = 0.03) and history of re-irradiation (Reirradiation vs. non-reirradiation: 9.7% vs. 8.1%, AHR = 5.95, 95% CI 1.25–28.29, *p* = 0.01) were both associated with higher risks of significant CAS in the Cox regression model (Appendix A). 

When the non-NPC group was further divided into the oral cavity/oropharyngeal, hypopharyngeal and laryngeal cancer subgroups, patients in the subgroups were all associated with higher risks of significant CAS than the NPC group (*p* < 0.001, respectively; Figure 3). However, the risk of developing significant CAS was similar between the oral cavity/oropharyngeal and hypopharyngeal cancers (log-rank *p* = 0.08), the oral cavity/oropharyngeal and laryngeal cancers (log-rank *p* = 0.12), as well as the hypopharyngeal and laryngeal cancer subgroups (log-rank *p* = 0.53).

The frequency of significant VA stenosis (non-NPC vs. NPC: 1.7% vs. 1.0%, *p* = 0.71), ICAS (non-NPC vs. NPC: 2.4% vs. 3.5%, *p* = 0.47), and IS (non-NPC vs. NPC: 1.0% vs. 0.3%, *p* = 0.57) were similar between the two groups. All the patients with significant CAS did not have pre-existing significant CAS before RT, were asymptomatic, and received medication treatment only. The only two patients of the non-NPC group underwent carotid artery interventions due to carotid blow-out syndrome (Table 2).

## 4. Discussion

Our study revealed that patients with non-NPC HNC were associated with a higher risk of significant CAS diagnosis within 5 years after RT completion. Scheduled carotid artery surveillance and monitoring of common vascular risk factors should be initiated earlier in patients with non-NPC HNC. By contrast, vascular surveillance for NPC patients could be deferred to 5 years after RT completion. Radiation vasculopathy usually develops 2–3 years after RT [21], and approximately 21% of survivors develop CAS within 5 years of RT completion [9]. Intervals of >5 years after RT completion, multiple vascular risk factors, NPC and laryngeal cancer have been determined to be independent predictors of significant CAS [2,9]. Previous guidelines suggested that carotid artery screening could be conducted 10 years after RT for HNC [19], while some studies have suggested screening for CAS in patients with HNC 2–5 years after RT [2]. However, consensus regarding the follow-up strategy, particularly for different HNC types, is yet to be reached [19]. RT for various HNC types has advanced considerably and may have thereby reduced the adverse effects on bystander organs [10]. Moreover, these advancements and their potential anti-atherosclerotic effects in patients with vascular risk factors may increase the importance of early detection of radiation vasculopathy. The development of a strategy for earlier and more precise vascular surveillance of patients with HNC nowadays is warranted, and our findings may serve as a reference in the development process.

Different types of HNC in this study may represent patients the patient population with various treatment methods, vascular risk factors, co-morbidities, diverse lifestyles, and different socio-economic composition. In our study, the patients with NPC had lower frequencies of smoking, and betel quid chewing than did the patients with non-NPC HNC. Smoking is a strong risk factor for NPC, laryngeal, and oral cavity cancers [22,23,24]. Diabetes, smoking, and hypertension are common risk factors for increased carotid plaque and carotid intima-media thickness [25]. Betel quid chewing is also a risk factor for atherosclerosis [26]. These factors may partly explain why the patients with NPC had a lower frequency of significant CAS during the follow-up period than those with non-NPC HNC did. Studies have also demonstrated that hypertension is associated with the risk of oral cavity and laryngeal cancer [27], and metabolic syndromes are significantly associated with the risk of laryngeal cancer [28]. By contrast, no overall association was identified between metabolic disorders and NPC [29], except for a previous study that reported an inverse association between diabetes and NPC [30]. Our report demonstrated a positive association between the glycated hemoglobin level and the risk of significant CAS development in Cox regression model. The non-NPC HNC patients had lower level of LDL at enrolment. It is possible that the non-NPC HNC patients, particularly those with oral cavity cancer, could have malnutrition after treatment [31]. After adjusting for these common vascular risk factors, patients of the non-NPC group remained associated with a higher risk of CAS diagnosis, suggesting the pathogenesis leading to the higher CAS risk in non-NPC could be more complex. Further, treatment methods could be different among these HNC patients. Oral cavity cancer patients mainly receive surgical treatment followed by post-operative RT while NPC patients majorly receive definitive RT. It is known that RT alone has higher risks of CAS than surgical treatment in HNC patients [4]. Surgical treatment in addition to RT could possibly injure carotid arteries [32]. Drug used in chemotherapy may also lead to vascular toxicity [33]. However, this may have minor influence on our results due to most of the HNC patients receiving platinum-based concurrent chemoradiation therapy in our hospital. Moreover, the radiation doses to carotid arteries may have impacts on the CAS development [2,34]. Without the dosimetric data of carotid arteries, our results were insufficient to discuss whether the difference of CAS risks between the two groups was associated with diverse radiation doses to carotid arteries. In general. the associated outcomes in this study could be related to the combination of multiple factors. Further studies may be needed to discuss underlying mechanisms.

In the present study, VA stenosis was observed less frequently than in previous studies [8]. PBT and VMAT may effectively reduce the radiation dose to the VA [10,35]. However, our results were insufficient to demonstrate the impacts of the advancement in RT methods on VA stenosis. ICAS was also reported to be less common among patients with radiation vasculopathy [8]. However, another study reported that ICAS could be at least as frequent as extracranial diseases in patients with radiation-induced stroke [36]. Patients with brain tumors may be at a higher risk of ICAS after undergoing IMRT or PBT [37]. The area irradiated during RT for NPC is more approximate to the intracranial vessels than that irradiated during RT for other sites of head and neck cancer. In the future, studies with larger sample sizes are required to evaluate the effect of RT on the risk of VA stenosis and ICAS in patients with HNC after RT.

Our results also showed that re-irradiation is associated with a higher risk of significant CAS development. This echoed the study showing an association between cumulative dose to organs at risk and the risk of developing carotid blowout syndrome [38]. PBT emerges as having a role for radiotherapy in the treatment of NPC more so than in the treatment of oral cavity or laryngeal cancer [39], and nearly half of the patients with NPC in the present study underwent PBT. In patients with NPC, the acute toxicity burden of PBT for nonmetastatic NPC was significantly lower than that of IMRT [40]. In the treatment of patients with early glottic cancer, PBT achieved superior outcomes for all organs at risk, including the contralateral carotid artery [41]. PBT also induced less toxicity in patients with ipsilateral salivary gland, oral–pharyngeal, or pharyngeal cancer [42,43,44]. Nevertheless, the effect of PBT on each individual carotid artery remains uncertain. It would be important to discuss the association between the RT method and CAS development. In the present study, methods of RT (PBT or VMAT) were not associated with a higher risk of significant CAS in our data (AHR = 1.30, 95% CI: 0.39–4.38, *p* = 0.67; Appendix A); however, our results could be limited due to the lack of comprehensive dosimetric data in the study. Further clinical trials are warranted to discuss this issue.

Our study has some limitations. First, the number of enrolled patients may have been too small to demonstrate clinical significance. However, our hospital is one of the largest medical centers in south-east Asia and China capable of providing all-course radiotherapy using proton therapy, and our study could be one of the largest studies on the topic conducted in this area especially for endemic NPC to date. In the future, a multicenter prospective study may be conducted to evaluate the accuracy of our conclusions. Second, associated vascular risk factors may affect clinicians’ decisions. The use of and compliance with prescribed medications for these vascular risk factors may have confounded the study results. A prospective study with a standardized medication protocol would more accurately identify differences in outcomes among various cancer types and RT methods. Third, the contrast-enhanced CT or MRI studies were reviewed to screen for secondary outcomes and pre-existed vascular lesions prior to RT. The timing of contrast injection was less suitable for vascular evaluation, which could have underpowered the results of vascular evaluation and could be a source of bias. A prospective study may help to validate our conclusions in the future. Fourth, we reviewed the data regarding details of RT retrospectively and therefore the dosimetric data to carotid arteries may not be available. This may have vital impact on discussing the casual relationship between CAS and types of HNC. However, the aim of this research was not to discuss the pathogenesis why non-NPC HNC patients are vulnerable to CAS development, but to provide a more precise clinical follow-up strategy of vascular complication between different HNCs. Fifth, there were a wide range of confounders regarding the presented data. Our conclusions might be at high risk of selection and reporting bias due to the large number of influencing factors and missing data as well as the large inhomogeneity of the patient cohort. The major factor of inhomogeneity is the above-mentioned issue of imaging disparities. Finally, the generalizability of our results to patients of other ethnicities remains to be explored. Due to the above limitations, these results were only hypothesis-generating.

## 5. Conclusions

Significant CAS within 5 years of RT completion was noted more frequently among the patients with non-NPC HNC than among those with NPC. Scheduled carotid artery surveillance and vascular risk monitoring should be commenced earlier for patients with non-NPC HNC. By contrast, vascular surveillance could be deferred to 5 years after RT completion for patients with NPC. Our findings may help guide the development of a more precise vascular follow-up strategy in the early years after RT.

## Figures and Tables

**Figure 1 jpm-12-01060-f001:**
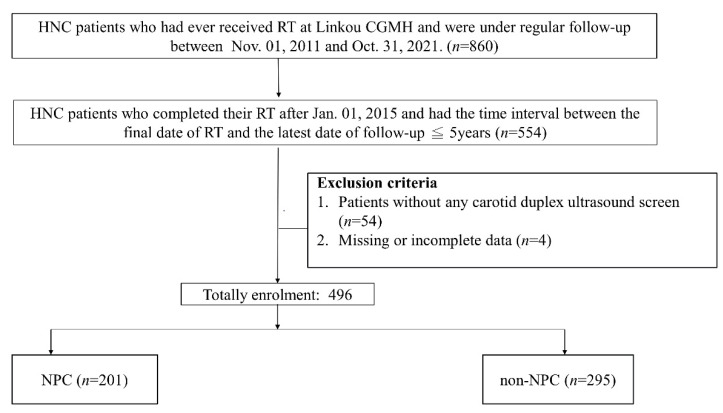
Patient enrollment. CGMH—Chang Gung Memorial hospital; HNC—head and neck cancer; NPC—nasopharyngeal carcinoma; RT—radiotherapy.

**Figure 2 jpm-12-01060-f002:**
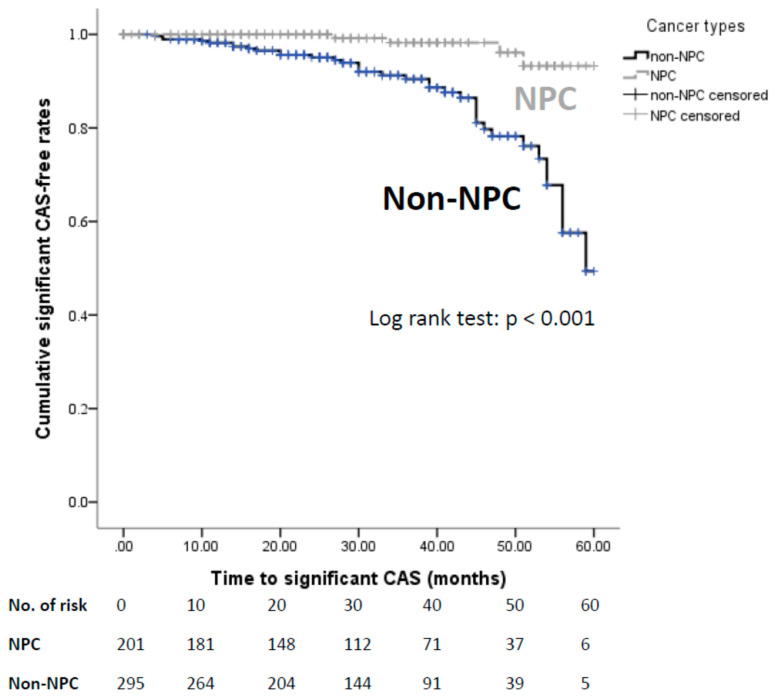
Cumulative significant CAS-free rates of patients with NPC and non-NPC head and neck cancer. Kaplan–Meier analysis comparing the significant CAS-free rates of the NPC and non-NPC groups. Frequency of significant CAS was higher in the non-NPC group than in the NPC group. CAS—carotid artery stenosis; NPC—nasopharyngeal carcinoma.

**Figure 3 jpm-12-01060-f003:**
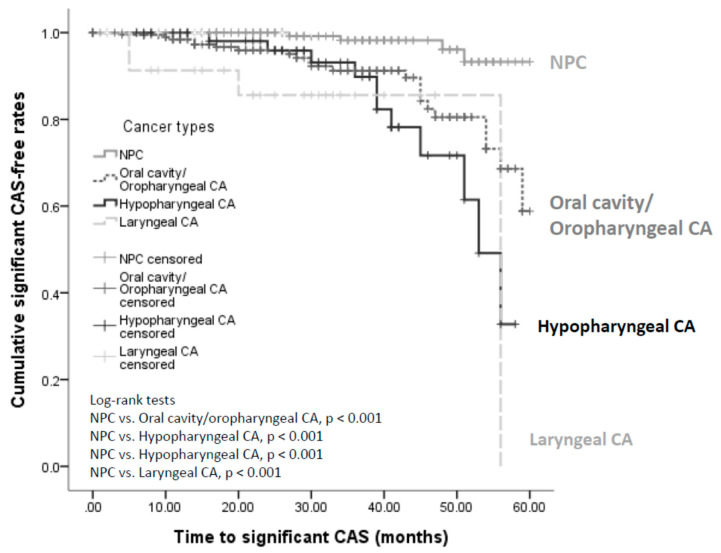
Cumulative significant CAS-free rates of patients with NPC, oral cavity/oropharyngeal CA, hypopharyngeal, and laryngeal CAs. Kaplan–Meier analysis comparing the significant CAS-free rates of patients with NPC, oral cavity/oropharyngeal, hypopharyngeal, and laryngeal CAs. Patients with oral cavity/oropharyngeal, hypopharyngeal, and laryngeal CA were more likely to develop significant CAS than were patients with NPC. However, the cumulative significant CAS-free rates were similar among oral cavity/oropharyngeal, hypopharyngeal, and laryngeal CAs. CA—cancer; CAS—carotid artery stenosis; NPC—nasopharyngeal carcinoma.

**Table 1 jpm-12-01060-t001:** Baseline characteristics of the study groups.

	NPC (*n* = 201)	Non-NPC (*n* = 295)	*p*
Demographics			
Age (years)	50.01 ± 10.84	58.42 ± 9.74	<0.01 ^†^
Sex (% male)	165 (82.5%)	269 (91.2%)	<0.01 ^†^
Hypertension (%)	28 (13.9%)	58 (19.9%)	0.08
Diabetes mellitus (%)	17 (8.5%)	34 (11.6%)	0.41
Dyslipidemia (%)	34 (16.9%)	46 (15.8%)	0.73
Coronary artery disease (%)	4 (2.0%)	8 (2.7%)	0.60
Chronic kidney disease (%)	1 (0.5%)	1 (0.3%)	0.91
Previous ischemic stroke (%)	0 (0.0%)	3 (1.0%)	0.27
Smoking (%)	77 (38.3%)	166 (56.8%)	<0.01 ^†^
Betel quid chewing (%)	28 (14.0%)	125 (42.8%)	<0.01 ^†^
RT dose (centigrays)	6808.82 ± 997.26	6624.07 ± 688.90	0.02 ^†^
Proton beam therapy (%)	85 (42.3%)	47 (15.9%)	<0.01 ^†^
Re-irradiation	12 (6.0%	19 (6.4%)	0.84
Cancer types			-
NPC	201 (100%)	0 (0%)	
Oral cavity/oropharyngeal cancer	0 (0%)	206 (69.8%)	
Laryngeal cancer	0 (0%)	23 (7.8%)	
Hypopharyngeal cancer	0 (0%)	59 (20.0%)	
Others	0 (0%)	7 (2.4%)	
Laboratory data			
Glycated hemoglobin (%)	5.66 ± 0.55	6.29 ± 0.58	0.29
Creatinine (mg/dL)	1.02 ± 1.08	1.04 ± 1.02	0.92
LDL (mg/dL)	125.54 ± 44.50	109.84 ± 44.66	<0.01 ^†^
HDL (mg/dL)	54.81 ± 16.08	49.55 ± 15.39	<0.01 ^†^
Triglyceride (mg/dL)	128.52 ± 93.58	147.54 ± 97.73	0.05
Free T4 (ng/dL)	1.25 ± 3.71	1.35 ± 6.67	0.86
CDU data at enrollment			
Mean IMT (left), mm	0.93 ± 2.84	0.82 ± 0.42	0.54
Mean IMT (right), mm	0.68 ± 0.24	0.76 ± 0.36	0.01 ^†^
Total plaque scores	1.21 ± 2.48	2.73 ± 4.13	<0.01 ^†^
Time from RT to the first CDU (months)	22.82 ± 14.35	21.79 ± 14.14	0.44
Time from RT to the last vascular follow-up (months)	33.16 ± 15.34	31.57 ± 16.37	0.28

CDU—carotid duplex ultrasound; HDL—high-density lipoprotein; IMT—intimal medial thickness; LDL—low-density lipoprotein; NPC—nasopharyngeal carcinoma; RT—radiation therapy; ^†^
*p* < 0.05. Data were examined by two-sample *t*-tests (continuous variables) and chi-square tests (categorical variables).

**Table 2 jpm-12-01060-t002:** Primary and secondary outcomes of this study.

Outcome	NPC *n* = 201 (%)	Non-NPC *n* = 295 (%)	NPC vs. Non-NPC	
			Adjusted HR (95% CI)	*p*-Value
Primary analysis ^#^				
Diagnosis of significant CAS	4 (2.0)	37 (12.7)	0.17 (0.05, 0.57)	0.004 ^†^
Secondary outcomes *				
Significant VA stenosis	2 (1.0)	5 (1.7)		0.71
ICAS	7 (3.5)	7 (2.4)		0.47
Ischemic stroke	2 (1.0)	1 (0.3)		0.57
Carotid artery interventions	0 (0.0)	2 (0.7)		0.52

CAS—carotid artery stenosis; CI—confidence interval; HR—hazard ratio; ICAS—intracranial artery stenosis; NPC—nasopharyngeal carcinoma; VA—vertebral artery; ^†^
*p* < 0.05. ^#^ Data was analyzed by multivariable Cox regression model with backward selection (Variables: NPC, smoking, diabetes mellitus, dyslipidemia, hypertension, coronary artery disease, glycated hemoglobin, RT doses, types of RT, and re-irradiation). * Data was examined by chi-square tests.

## Data Availability

Data available on request due to privacy and ethical restrictions.

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
