# Peer review of "Head and Neck Cancer Types and Risks of Cervical–Cranial Vascular Complications within 5 Years after Radiation Therapy"

_jpm, 2022, doi:10.3390/jpm12071060_

Round 1

Reviewer 1 Report

Summary:

The study evaluates the frequency of late vasculopathy of the carotid artery in 496 patients who underwent primary radiotherapy/chemoradiation.

The title is conclusive and precise.

The language is appropriate but needs revision by a native speaker.

Funding of the study and a positive review board decision are provided.

Detailed review:

Originality/Novelty: The authors have clearly defined the study objectives.

The presented data provide new insights of late vascular toxicities of primary RT. The radiation oncology community will benefit from the results of this analysis. The design of future trials could be influenced by the presented data. The issue of the paper is of interest for readers and will attract a wide readership.

Significance:

In order to improve readability, the authors should consider describing inclusion criteria as precise as possible. They report on reviewed patient data from November 1, 2011, but in the next sentence they point out that patients treated before 2015 were excluded from analysis. The first figure should be corrected accordingly (there is not number provided for the first figure, the trial profile, which should be corrected as well).

A clear definition of the aims of the study is provided.

The conclusions (CAS surveillance and monitoring of risk factors for non-NPC patients, deferring CAS surveillance to 5 years in NPC patients, etc.) are not supported by the presented data. VA stenosis was not significantly different in PBT and VMAT patients (page 15 last line). Yet, this is no results that can be considered consistent with a study showing that RT dose is lower in PBT compared with VMAT, as it reflects a different issue. The authors should provide dosimetric data to allow for this assumption.

In the discussion section the authors should also comment on the fact that non-NPC patients had significantly lower LDL levels, which is usually associated with lower risk of atherosclerosis.

The discussion of the advantages of PBT is not helpful for interpretation of the presented data as there is no difference among the two treatment types.

There is a wide range of confounders regarding the presented data. The authors should consider acknowledging that their drawn conclusions are at high risk of selection and reporting bias due to the large number of influencing factors and missing data as well as the large inhomogeneity of the patient cohort. The major factor of inhomogeneity is the above-mentioned issue of imaging disparities.

The authors should consider pointing out that these results are only hypothesis generating. The paper would gain substantially for instance by providing a proposal for a prospective study, especially by including proposals on stratification factors.

The major shortcoming of the paper is the lack of a precise and thorough analysis of limitations (e. g. no randomized comparison, large variance of factors influencing the results, imaging insufficiencies (see statement above), and so forth). The mentioned shortcomings should be described in detail and elaborated thoroughly.

Quality of Presentation:

The structure of the article is written in an appropriate way.

In the follow-up section the authors should provide information on the frequency of serial carotid duplex ultrasound (CDU) examinations. Was CDU performed on every follow-up and was it performed in all patients?

How was assured that investigators performed duplex sonography appropriately? Is there a specific curriculum? Has teaching been performed at an accredited teaching facility? Was there an examination of ultrasound personnel? (e. g. Position statement and best practice recommendations on the imaging use of ultrasound from the European Society of Radiology ultrasound subcommittee. Insights Imaging 11, 115 (2020). https://doi.org/10.1186/s13244-020-00919-x).

The authors should consider giving a statement on standards in diagnosing CAS (e.g. Jonas DE, Feltner C, Amick HR, et al. Screening for asymptomatic carotid artery stenosis: a systematic review and meta-analysis for the US Preventive Services Task Force. Ann Intern Med. 2014;161(5):336-346.). CT angiography but not normal CT is considered standard to evaluate CAS.

There are several data missing in the patient characteristics that may be considered relevant for interpretation of results. For instance, the respective numbers of patients who had information on CAS before RT, the type of imaging in these patients, the rate of CAS in patients with re-irradiation, the number of patients who underwent CDU, MRI and CT for diagnosis of CAS, etc. The authors should consider providing more detailed information.

The work will provide an advance towards the current knowledge, but it must be thoroughly revised.

Author Response

# Reviewer 1

Dear Reviewer

Thank you for your detailed review. Your thorough and excellent suggestions not only made our manuscript better but also guided us to improve the quality of our ongoing cohort data.

 We have provided a point-by-point revision and our responses to all your comments. The reasons and revisions are provided below. In the revised manuscript, all the changes are highlighted. We deeply appreciate your valuable review, which stimulated a more thorough consideration of the article. Thank you very much and we hope the revised manuscript became much better and reached the standard of JPM.

Sincerely Yours,

Yeu-Jhy Chang, MD

Stroke Center and Department of Neurology

Chang Gung Memorial Hospital, Linkou Medical Center and College of Medicine, Chang Gung University, Taoyuan, Taiwan

No. 5, Fu-Hsing ST. Kueishan, Taoyuan, 33333 Taiwan

Tel: 886-3-3281200 ext 8340

And 

Joseph Tung-Chieh Chang MD, MHA

Department of Radiation Oncology, Proton and Radiation Therapy Center, Chang Gung Medical Foundation, Linkou Chang

Gung Memorial Hospital, Taoyuan, Taiwan.

  1. The study evaluates the frequency of late vasculopathy of the carotid artery in 496 patients who underwent primary radiotherapy/chemoradiation. The title is conclusive and precise.The language is appropriate but needs revision by a native speaker. Funding of the study and a positive review board decision are provided.

 Answer: thank you for your comment.

  1. Originality/Novelty:The authors have clearly defined the study objectives.The presented data provide new insights of late vascular toxicities of primary RT. The radiation oncology community will benefit from the results of this analysis. The design of future trials could be influenced by the presented data. The issue of the paper is of interest for readers and will attract a wide readership.

 Answer: thank you for your comment.

  1. In order to improve readability, the authors should consider describing inclusion criteria as precise as possible. They report on reviewed patient data from November 1, 2011, but in the next sentence they point out that patients treated before 2015 were excluded from analysis. The first figure should be corrected accordingly (there is not number provided for the first figure, the trial profile, which should be corrected as well).

Answer: thank you for your suggestion, we revised the figure 1 to show our patient recruitment protocol. We hoped this may reduce the confusions.

Method, page 4, line 11-20

Patient recruitment and demographic data

Our cohort included patients with HNC who had ever undergone RT between November 1, 2011, and October 31, 2021, and who received regular follow-ups at the radio-oncology and neurology departments of Chang Gung Memorial Hospital at Linkou. Because we aimed to investigate neurovascular complications during the early phase after RT and hope to minimize the bias caused by the variance of RT methods in different era. We retrospectively reviewed the patients completed their RT after January 01, 2015 in this study, who also had the time interval between the final date of RT and the latest date of follow-up ≦ 5years. Patients who had missing data or had never received carotid duplex ultrasound (CDU) after RT were further excluded.

  1. A clear definition of the aims of the study is provided.

 Answer: thank you, we revised the manuscript to make our study aim much clear.

Introduction, page 4, line 6-8

However, the incidence of cervical–cranial vascular complications in the patients undergoing these advanced RT methods remain unknown. In the present study, we investigated the frequency of cervical–cranial vascular complications among patients with various HNC types in current era and hoped to develop a precise clinical follow-up strategy of vascular complication between different HNC types.

  1. The conclusions (CAS surveillance and monitoring of risk factors for non-NPC patients, deferring CAS surveillance to 5 years in NPC patients, etc.) are not supported by the presented data. VA stenosis was not significantly different in PBT and VMAT patients (page 15 last line). Yet, this is no results that can be considered consistent with a study showing that RT dose is lower in PBT compared with VMAT, as it reflects a different issue. The authors should provide dosimetric data to allow for this assumption.

Answer: we are sorry for making this confusion, we’ve revised the manuscript accordingly.

Discussion, page 17, line 5-8

In the present study, VA stenosis was observed less frequently than in previous studies [8]. PBT and VMAT, may effectively reduce the radiation dose to the VA [10, 35]. However, our results were insufficient to demonstrate the impacts of the advancement in RT methods on VA stenosis. In the future, studies with larger sample sizes should be conducted to examine this difference.

Lack of dosimetric data could be a weakness particularly when the aim of the study was to discuss the casual relationship between RT method and CAS development, or the underlying mechanism of RT related CAS. However, the major objective of this study was to propose a useful and more precise vascular follow-up strategy in these HNC patients after RT. We hoped to alert the clinician to beware of the higher risks of significant CAS in the non-NPC patients, even in early years after RT. Non-NPC and NPC groups may represent the patients received different treatment method, associated vascular risk factors, radiation doses to carotid arteries, and socioeconomic status. Types of HNC would be a practical clinical marker to select the vulnerable patients, since doses to carotid arteries are not easily acquired for most of the primary clinician, especially for non-radio-oncologists.

We revised our manuscript to address this limitation.

Discussion section/ limitations.

Page 15, line 21-23

Different types of HNC in this study may represent patients the patient population with various treatment methods, vascular risk factors, co-morbidities, diverse life styles, and different socio-economic composition.

Page 16, line 24 – page 17, line 4

Moreover, the radiation doses to carotid arteries may have impacts on the CAS development [2, 34]. Without the dosimetric data of carotid arteries, our results were insufficient to discuss whether the difference of CAS risks between the two groups was associated with diverse radiation doses to carotid arteries. In general. the associated outcomes in this study could be related to the combination of multiple factors. Further studies maybe needed to discuss underlying mechanisms.

Page 18, line 4-8

In the present study, method of RT (PBT or VMAT) was not associated with a higher risk of significant CAS in our data (AHR = 1.30, 95% CI: 0.39-4.38, p = 0.67; supplementary table), however our results could be limited due to the lack of comprehensive dosimetric data in the study. Further clinical trials are warranted to discuss this issue. 

Page 18, line 23 – page 18, line 4

Fourth, we reviewed the data regarding details of RT retrospectively and therefore the dosimetric data to carotid arteries may not be available. This may have vital impact on discussing the casual relationship between CAS and types of HNC. However, the aim of this research was not to discuss the pathogenesis why non-HNC patients are vulnerable to CAS development but was to provide a more precise clinical follow-up strategy of vascular complication between different HNCs.

  1. In the discussion section the authors should also comment on the fact that non-NPC patients had significantly lower LDL levels, which is usually associated with lower risk of atherosclerosis.

Answer: thank you for your comment, we revised the manuscript accordingly.

Discussion, page 16, line 12-16

The non-NPC patients had lower level of LDL at enrolment. It is possible that the non-NPC, particular oral cavity cancer, patients could have malnutrition after treatment [31]. After adjusting for these common vascular risk factors, patients of non-NPC remained associated with a higher risk of CAS diagnosis, suggesting the pathogenesis leading to the higher CAS risk in non-NPC could be more complex.

  1. The discussion of the advantages of PBT is not helpful for interpretation of the presented data as there is no difference among the two treatment types.

Answer: thank you for your suggestion, we agreed with your opinion. We’ve shorten the discussion regarding the discussion regarding PBT and its advantages, and mentioned the limitation why our results were insufficient to give a conclusive answer.

Discussion, page 17, line 20 – page 18, line 8

PBT emerges it role for radiotherapy in the treatment of NPC than in the treatment of oral cavity or laryngeal cancer [39], and nearly half of the patients with NPC in the present study underwent PBT. In patients with NPC, the acute toxicity burden of PBT for nonmetastatic NPC was significantly lower than that of IMRT [40]. In the treatment of patients with early glottic cancer, PBT achieved superior outcomes for all organs at risk, including the contralateral carotid artery [41]. PBT also induced less toxicity in patients with ipsilateral salivary gland, oral–pharyngeal, or pharyngeal cancer [42-44]. Nevertheless, the effect of PBT on each individual carotid artery remains uncertain. It would be important to discuss the association between the RT method and CAS development. In the present study, method of RT (PBT or VMAT) was not associated with a higher risk of significant CAS in our data (AHR = 1.30, 95% CI: 0.39-4.38, p = 0.67; supplementary table), however our results could be limited due to the lack of comprehensive dosimetric data in the study. Further clinical trials are warranted to discuss this issue. 

  1. There is a wide range of confounders regarding the presented data. The authors should consider acknowledging that their drawn conclusions are at high risk of selection and reporting bias due to the large number of influencing factors and missing data as well as the large inhomogeneity of the patient cohort. The major factor of inhomogeneity is the above-mentioned issue of imaging disparities.

Answer: thank you for your comment, we revised the limitation session accordingly.

Discussion, page 19, line 5-8

Our conclusions might be at high risk of selection and reporting bias due to the large number of influencing factors and missing data as well as the large inhomogeneity of the patient cohort. The major factor of inhomogeneity is the above-mentioned issue of imaging disparities.

  1. The authors should consider pointing out that these results are only hypothesis generating. The paper would gain substantially for instance by providing a proposal for a prospective study, especially by including proposals on stratification factors.

Answer: thank you for your comment, we revised the limitation session accordingly.

Discussion, page 19, line 8-12

Finally, the generalizability of our results to patients of other ethnicities remains to explore. Due to the above limitations, these results were only hypothesis generating. The paper would gain substantially for instance by providing a proposal for a prospective study, especially by including proposals on stratification factors.

  1. The major shortcoming of the paper is the lack of a precise and thorough analysis of limitations (e. g. no randomized comparison, large variance of factors influencing the results, imaging insufficiencies (see statement above), and so forth). The mentioned shortcomings should be described in detail and elaborated thoroughly.

Answer: thank you for your comment, we revised the limitation session accordingly.

Discussion, page 18, line 9- page 9, line 12

Our study has some limitations. First, the number of enrolled patients may have been too small to demonstrate clinical significance. However, our hospital is one of the large medical centers in south-east Asia and China capable of providing all course radiotherapy using proton therapy, and our study could be one of the large studies on the topic conducted in this area especially for endemic NPC to date. In the future, a multicenter prospective study may be conducted to evaluate the accuracy of our conclusions. Second, associated vascular risk factors may affect clinicians’ decisions. The use of and compliance with prescribed medications for these vascular risk factors may have confounded the study results. A prospective study with a standardized medication protocol would more accurately identify differences in outcomes among various cancer types and RT methods. Third, the contrast enhanced CT or MRI studies were reviewed to screen for secondary outcomes and pre-existed vascular lesions prior to RT. The timing of contrast injection was less suitable for vascular evaluation, which could have underpowered the results of vascular evaluation and could be a source of bias. A prospective study may help to validate our conclusions in the future. Fourth, we reviewed the data regarding details of RT retrospectively and therefore the dosimetric data to carotid arteries may not be available. This may have vital impact on discussing the casual relationship between CAS and types of HNC. However, the aim of this research was not to discuss the pathogenesis why non-HNC patients are vulnerable to CAS development but was to provide a more precise clinical follow-up strategy of vascular complication between different HNCs. Fifth, there were a wide range of confounders regarding the presented data. Our conclusions might be at high risk of selection and reporting bias due to the large number of influencing factors and missing data as well as the large inhomogeneity of the patient cohort. The major factor of inhomogeneity is the above-mentioned issue of imaging disparities. Finally, the generalizability of our results to patients of other ethnicities remains to explore. Due to the above limitations, these results were only hypothesis generating. The paper would gain substantially for instance by providing a proposal for a prospective study, especially by including proposals on stratification factors.

  1. In the follow-up section the authors should provide information on the frequency of serial carotid duplex ultrasound (CDU) examinations. Was CDU performed on every follow-up and was it performed in all patients? How was assured that investigators performed duplex sonography appropriately? Is there a specific curriculum? Has teaching been performed at an accredited teaching facility? Was there an examination of ultrasound personnel? (e. g. Position statement and best practice recommendations on the imaging use of ultrasound from the European Society of Radiology ultrasound subcommittee. Insights Imaging 11, 115 (2020). https://doi.org/10.1186/s13244-020-00919-x).The authors should consider giving a statement on standards in diagnosing CAS (e.g. Jonas DE, Feltner C, Amick HR, et al. Screening for asymptomatic carotid artery stenosis: a systematic review and meta-analysis for the US Preventive Services Task Force. Ann Intern Med. 2014;161(5):336-346.). CT angiography but not normal CT is considered standard to evaluate CAS.

Answer: thank you for your suggestion, that was indeed an important question.

Please also allow me to introduce how these data was recruited.

Our institution (Chang Gung memorial hospital) might be the largest center treating head and neck cancer in Taiwan. The two corresponding authors (Professor TC Chang and YJ Chang) had noticed the high proportion of carotid artery stenosis after RT in 2009 and presented the first manuscript regarding this issue (J Vasc Surg. 2009;50:280–5.) Thereafter, Professor TC Chang referred some of his post-RT patients to receive vascular surveillance.

In late 2015, the proton therapy center was set-up in our institution. In order to provide more information for the share decision management process of RT, we reviewed and analyzed the data collected between 2013 and 2014, and therefore we had the two another related manuscript (BMC Neurol 2021;21(1):30. and PLoS One 2021;16(2):e0246684.). After that, more and more radio-oncologists in our institution noticed this issue and referred most of their patients for vascular surveillance after RT (for clinical practice, not for study). The three co-authors (Prof. TC Chang, CY Lin, and BS Huang) treated more then 90% of the head and neck cancer patients in our institution and that’s why was have such number of patients’ data in these years.

How were these patients followed?

Generally, these patients received the same principle of vascular surveillance clinical in our neurology department.

  1. Our radio-oncologists did not perform the carotid artery screen before RT. According to the guideline for screen of extracranial carotid artery disease (J Neuroimaging. 2007;17(1):19-47.), there’s inconclusive data to support this vascular screen. In Taiwan, medical fees are paid from our national health insurance bureau only when such treatment is “fulfilled with the guideline suggestions”.
  2. Our radio-oncologists referred most of their patients 1-3 years after RT completion (according to previous literature, the post-RT carotid stenosis could be noted 3-5 years after the treatment) regardless their head and neck cancer types.
  3. Upon visit in Neurology department, we mainly performed first vascular surveillance by means of carotid ultrasound due to (1) Carotid ultrasound is an easy, non-invasive screen tool with proven good sensitivity to carotid stenosis in our institution (Angiology 2015; 66:180-6.); (2) These patients are already received contrast enhanced MRI (not MRA) and/or CT (not CTA) for their cancer recurrence or metastasis follow-up, we think we should reduce their contrast and radiation exposures.
  4. We usually repeated the next carotid ultrasound 12-24 months ((J Neuroimaging. 2007;17(1):19-47.) based on patients’ carotid status, vascular risk factors, and cancer types (after we analyzed the data in this manuscript). Of the patients without significant carotid artery stenosis (i.e < 50% at ICA or CCA), we may repeat carotid ultrasound 12-24 months again after the follow-up study.
  5. For the patients with possible significant carotid artery stenosis (i.e > 50% at ICA or CCA) on carotid ultrasound, we arranged CT angiography or MR angiography to “”confirm”” the diagnosis. We also started to control the vascular risk factors of these patients more strictly and may consider antiplatelet treatment or carotid artery stenting in them. That’s why we need to confirm the diagnosis before beginning of these treatments. We knew that we did not have the pre-RT vascular imaging, and this may bring a question “ whether these patients already had significant carotid artery stenosis prior to RT?”. Therefore, we reviewed each source images and axial cuts of the head and neck CT/MRI performed for the “cancer staging” before RT. If they had existed significant (>50%) carotid artery stenosis before RT, we could find it from these images. Of the patients with significant carotid artery stenosis in this data, “NO” patients had significant carotid stenosis before RT after the retrospective image review.

In order to describe the follow-up strategy and how we tried to exclude the existing carotid artery stenosis prior to RT, we revised the outcome and follow-up paragraph of the METHOD section.

Methods, Page 6, line 22 – page 8, line 4

Outcomes and follow-ups

The primary outcome in this study was the diagnosis of significant CAS within the first 5 years after RT. We defined significant CAS as >50% stenosis on B-mode images with a peak systolic velocity ≥120 cm/s based on the hemodynamic criteria for any internal or common carotid artery in the CDU examination. The degree of CAS was determined according to the standard ultrasound criteria [17]. Personnel from our CDU laboratory diagnosed CAS with an overall accuracy of >90% [18]. Routine CDU evaluation prior to the initiation of RT was not recommended in previous guideline [19], we reviewed the images of carotid and vertebral arteries (VA) in the contrast enhanced CT and/or magnetic resonance imaging (MRI) performed before the initiation of cancer treatment to detect preexisted significant CAS or VA stenosis. We defined as a pre-existed significant CAS/ VA stenosis if there were any plaques/stenosis occupied more than 50% of the vessel lumen in the axial views. We hoped this could help to verify whether the significant CAS developed after RT or existed prior to RT.

The secondary outcomes were the presence of significant vertebral artery (VA) stenosis, significant intracranial arterial stenosis (ICAS), and occurrence of symptomatic ischemic stroke (IS). Types of carotid interventions were also recorded.

The patients received regular follow-up at the radio-oncology department at least every 6 months after RT. Contrast enhanced CT and/or MRI were arranged in the meantime to identify potential distal metastasis or recurrence of the primary tumor. To monitor the neurovascular complications, patients were also referred to our neurology out-patient department during the post-RT period. Upon the first visit at neurology department, the patients’ laboratory data including complete blood count, blood chemistry, lipid profile and thyroid hormone levels were checked. Meanwhile, patients underwent the first carotid duplex ultrasound (CDU) study to screen the presence of significant CAS. Total plaque scores (TPS) [6], intimal medial thickness (IMT) of the common carotid arteries, and degree of CAS were also measured in this examination. Patients received repeated CDU exam 12-24 months later based on the severity of carotid artery lesions in the first visit [19]. We also reviewed the patients’ CT/MR images arranged for cancer follow-ups to find the possible signs of secondary outcomes.

In patients noted with a sign of significant CAS on any CDU study, significant VA stenosis or ICAS, we then arranged CT angiography or MR angiography to confirm the diagnosis.

  1. There are several data missing in the patient characteristics that may be considered relevant for interpretation of results. For instance, the respective numbers of patients who had information on CAS before RT, the type of imaging in these patients, the rate of CAS in patients with re-irradiation, the number of patients who underwent CDU, MRI and CT for diagnosis of CAS, The authors should consider providing more detailed information.

 Answer: thank you for your comment.

  • None of the patients had CAS before RT. We reviewed the axial images of CT/MRI arranged for cancer staging before RT in patients diagnosed as having CAS during neurology follow-up.
  • All the patients received CDU, and for those with suspicious significant CAS, MRA or CTA were arranged to confirmed the diagnosis.
  • The frequency of CAS was 9.7% in patients with reirradiation, but 8.1% in patient without reirradiation.

We’ve provided this information in the methods section

Methods, Page 6, line 22 – page 8, line 4

Outcomes and follow-ups

The primary outcome in this study was the diagnosis of significant CAS within the first 5 years after RT. We defined significant CAS as >50% stenosis on B-mode images with a peak systolic velocity ≥120 cm/s based on the hemodynamic criteria for any internal or common carotid artery in the CDU examination. The degree of CAS was determined according to the standard ultrasound criteria [17]. Personnel from our CDU laboratory diagnosed CAS with an overall accuracy of >90% [18]. Routine CDU evaluation prior to the initiation of RT was not recommended in previous guideline [19], we reviewed the images of carotid and vertebral arteries (VA) in the contrast enhanced CT and/or magnetic resonance imaging (MRI) performed before the initiation of cancer treatment to detect preexisted significant CAS or VA stenosis. We defined as a pre-existed significant CAS/ VA stenosis if there were any plaques/stenosis occupied more than 50% of the vessel lumen in the axial views. We hoped this could help to verify whether the significant CAS developed after RT or existed prior to RT.

The secondary outcomes were the presence of significant vertebral artery (VA) stenosis, significant intracranial arterial stenosis (ICAS), and occurrence of symptomatic ischemic stroke (IS). Types of carotid interventions were also recorded.

The patients received regular follow-up at the radio-oncology department at least every 6 months after RT. Contrast enhanced CT and/or MRI were arranged in the meantime to identify potential distal metastasis or recurrence of the primary tumor. To monitor the neurovascular complications, patients were also referred to our neurology out-patient department during the post-RT period. Upon the first visit at neurology department, the patients’ laboratory data including complete blood count, blood chemistry, lipid profile and thyroid hormone levels were checked. Meanwhile, patients underwent the first carotid duplex ultrasound (CDU) study to screen the presence of significant CAS. Total plaque scores (TPS) [6], intimal medial thickness (IMT) of the common carotid arteries, and degree of CAS were also measured in this examination. Patients received repeated CDU exam 12-24 months later based on the severity of carotid artery lesions in the first visit [19]. We also reviewed the patients’ CT/MR images arranged for cancer follow-ups to find the possible signs of secondary outcomes.

In patients noted with a sign of significant CAS on any CDU study, significant VA stenosis or ICAS, we then arranged CT angiography or MR angiography to confirm the diagnosis.

And Results section

Results, page 9, line 21-23

The mean interval from the RT completion to the last follow up were similar between the two groups. None of the patients noted with significant CAS during follow-up had pre-existed lesion prior to RT.

Results, page 10, line 3-6

the glycated hemoglobin level (AHR = 1.03, 95% CI 1.01-1.06, p = 0.03) and history of re-irradiation (Reirradiation vs. non-reirradiation: 9.7% vs. 8.1%, AHR = 5.95, 95% CI 1.25-28.29, p = 0.01) were both associated with higher risks of significant CAS in the Cox-regression model (Supplementary table).

Reviewer 2 Report

Please see the attached file "Review report.pdf"

Author Response

# Reviewer 2

Dear Reviewer

Thank you for your detailed review. We are so glad to have the opportunity to have your review. Your thorough and excellent suggestions not only made our manuscript better but also guided us to improve the quality of our ongoing cohort data.

 We have provided a point-by-point revision and our responses to all your comments. The reasons and revisions are provided below. In the revised manuscript, all the changes are highlighted. We deeply appreciate your valuable review, which stimulated a more thorough consideration of the article. Thank you very much and we hope the revised manuscript became much better and reached the standard of JPM.

Sincerely Yours,

Yeu-Jhy Chang, MD

Stroke Center and Department of Neurology

Chang Gung Memorial Hospital, Linkou Medical Center and College of Medicine, Chang Gung University, Taoyuan, Taiwan

No. 5, Fu-Hsing ST. Kueishan, Taoyuan, 33333 Taiwan

Tel: 886-3-3281200 ext 8340

And 

Joseph Tung-Chieh Chang MD, MHA

Department of Radiation Oncology, Proton and Radiation Therapy Center, Chang Gung Medical Foundation, Linkou Chang

Gung Memorial Hospital, Taoyuan, Taiwan.

A brief summary

  1. The aim of the paper was to investigate the frequency of carotid artery stenosis (CAS), evaluated by carotid duplex ultrasound (CDU) after radiation therapy (RT) in head and neck cancer patients (HNC) and to identify differences among patients with nasopharyngeal (NPC) vs. non-NPC cancer. The main contribution of this paper to general knowledge is limited due to several methodological flaws, however, the overall number of included patients (n=496) is its relative strength.

The results probably include previously presented authors’ data1,2, however, the herein presented results are in general new and worthy of presentation.

Answer: Thank you for your review.

We’d like to explain that the data used in this manuscript was different from our previous presented data. The data used in our previous publications was recruited from the patients who completed their radiation therapy (RT) between 2013 and 2014. But the data used in this manuscript was recruited from the patients who completed their RT after 2015.

Please also allow me to introduce how these data was recruited.

Our institution (Chang Gung memorial hospital) might be the largest center treating head and neck cancer in Taiwan. The two corresponding authors (Professor TC Chang and YJ Chang) had noticed the high proportion of carotid artery stenosis after RT in 2009 and presented the first manuscript regarding this issue (J Vasc Surg. 2009;50:280–5.) Thereafter, Professor TC Chang referred some of his post-RT patients to receive vascular surveillance.

In late 2015, the proton therapy center was set-up in our institution. In order to provide more information for the share decision management process of RT, we reviewed and analyzed the data collected between 2013 and 2014, and therefore we had the two another related manuscript (BMC Neurol 2021;21(1):30. and PLoS One 2021;16(2):e0246684.). After that, more and more radio-oncologists in our institution noticed this issue and referred most of their patients for vascular surveillance after RT (for clinical practice, not for study). The three co-authors (Prof. TC Chang, CY Lin, and BS Huang) treated more than 90% of the head and neck cancer patients in our institution and that’s why we have such number of patients’ data in these years.

How were these patients followed?

Generally, these patients received the same principle of vascular surveillance clinical in our neurology department.

  1. Our radio-oncologists did not perform the carotid artery screen before RT. According to the guideline for screen of extracranial carotid artery disease (J Neuroimaging. 2007;17(1):19-47.), there’s inconclusive data to support this vascular screen. In Taiwan, medical fees are paid by our national health insurance bureau only when such management is “fulfilled with the guideline suggestions”.
  2. Our radio-oncologists referred most of their patients 1-3 years after RT completion (according to previous literature, the post-RT carotid stenosis could be noted 3-5 years after the treatment) regardless their head and neck cancer types.
  3. Upon visit in Neurology department, we mainly performed first vascular surveillance by means of carotid ultrasound due to (1) Carotid ultrasound is an easy, non-invasive screen tool with proven good sensitivity to carotid stenosis in our institution (Angiology 2015; 66:180-6.); (2) These patients were already received contrast enhanced MRI (not MRA) and/or CT (not CTA) for their cancer recurrence or metastasis follow-up, we hoped to reduce their contrast medium and radiation exposures.
  4. We usually repeated the next carotid ultrasound 12-24 months ((J Neuroimaging. 2007;17(1):19-47.) based on patients’ carotid status, vascular risk factors, and cancer types (after we analyzed the data in this manuscript). In the patients without significant carotid artery stenosis (i.e < 50% at ICA or CCA), we may repeat carotid ultrasound 12-24 months again for the follow-up study.
  5. For the patients with possible significant carotid artery stenosis (i.e > 50% at ICA or CCA) on carotid ultrasound, we arranged CT angiography or MR angiography to “”confirm”” the diagnosis. We also started to control the vascular risk factors of these patients more strictly and may consider antiplatelet treatment or carotid artery stenting in them. That’s why we needed to confirm the diagnosis before beginning of these treatments. We knew that we did not have the pre-RT vascular imaging, and this may bring a question “ whether these patients already had significant carotid artery stenosis prior to RT?”. Therefore, we reviewed each source images and axial cuts of the head and neck CT/MRI performed for the “cancer staging” before RT. If they had existed significant (>50%) carotid artery stenosis before RT, we could find it from these images. Among the patients with significant carotid artery stenosis in this report, “NO” patients had significant carotid stenosis before RT after the retrospective image review.

In order to describe the follow-up strategy and how we tried to exclude the existing carotid artery stenosis prior to RT, we revised the outcome and follow-up paragraph of the METHOD section.

Methods, Page 6, line 22 – page 8, line 4

Outcomes and follow-ups

The primary outcome in this study was the diagnosis of significant CAS within the first 5 years after RT. We defined significant CAS as >50% stenosis on B-mode images with a peak systolic velocity ≥120 cm/s based on the hemodynamic criteria for any internal or common carotid artery in the CDU examination. The degree of CAS was determined according to the standard ultrasound criteria [17]. Personnel from our CDU laboratory diagnosed CAS with an overall accuracy of >90% [18]. Routine CDU evaluation prior to the initiation of RT was not recommended in previous guideline [19], we reviewed the images of carotid and vertebral arteries (VA) in the contrast enhanced CT and/or magnetic resonance imaging (MRI) performed before the initiation of cancer treatment to detect preexisted significant CAS or VA stenosis. We defined as a pre-existed significant CAS/ VA stenosis if there were any plaques/stenosis occupied more than 50% of the vessel lumen in the axial views. We hoped this could help to verify whether the significant CAS developed after RT or existed prior to RT.

The secondary outcomes were the presence of significant vertebral artery (VA) stenosis, significant intracranial arterial stenosis (ICAS), and occurrence of symptomatic ischemic stroke (IS). Types of carotid interventions were also recorded.

The patients received regular follow-up at the radio-oncology department at least every 6 months after RT. Contrast enhanced CT and/or MRI were arranged in the meantime to identify potential distal metastasis or recurrence of the primary tumor. To monitor the neurovascular complications, patients were also referred to our neurology out-patient department during the post-RT period. Upon the first visit at neurology department, the patients’ laboratory data including complete blood count, blood chemistry, lipid profile and thyroid hormone levels were checked. Meanwhile, patients underwent the first carotid duplex ultrasound (CDU) study to screen the presence of significant CAS. Total plaque scores (TPS) [6], intimal medial thickness (IMT) of the common carotid arteries, and degree of CAS were also measured in this examination. Patients received repeated CDU exam 12-24 months later based on the severity of carotid artery lesions in the first visit [19]. We also reviewed the patients’ CT/MR images arranged for cancer follow-ups to find the possible signs of secondary outcomes.

In patients noted with a sign of significant CAS on any CDU study, significant VA stenosis or ICAS, we then arranged CT angiography or MR angiography to confirm the diagnosis.

  1. General concept comments

The manuscript’s subject is relevant for the field, however, there are important weaknesses of the study:

  • the lack of evaluation of CAS by CDU before RT, and

Answer: Thank you for your comment. We have explained the principles how we evaluated the vascular status before RT in previous responses.

For the patients with possible significant carotid artery stenosis on carotid ultrasound, we arranged CT angiography or MR angiography to “”confirm”” the diagnosis. We also started to control the vascular risk factors of these patients more strictly. We knew that we did not have the pre-RT vascular imaging, and this may bring a question “ whether these patients already had significant carotid artery stenosis prior to RT?”. Therefore, we reviewed each source images and axial cuts of the head and neck CT/MRI performed for the “cancer staging” before RT. If they had existed significant (>50%) carotid artery stenosis before RT, we could find from these images. Of the patients with significant carotid artery stenosis in this data, “NO” patients had significant carotid stenosis before RT after the retrospective image review.

In order to describe the follow-up strategy and how we tried to exclude the existing carotid artery stenosis prior to RT, we revised the outcome and follow-up paragraph of the METHOD and RESLUT section.

Methods, page 7, line 3-10

Routine CDU evaluation prior to the initiation of RT was not recommended in previous guideline [19], we reviewed the images of carotid and vertebral arteries (VA) in the contrast enhanced CT and/or magnetic resonance imaging (MRI) performed before the initiation of cancer treatment to detect preexisted significant CAS or VA stenosis. We defined as a pre-existed significant CAS/ VA stenosis if there were any plaques/stenosis occupied more than 50% of the vessel lumen in the axial views. We hoped this could help to verify whether the significant CAS developed after RT or existed prior to RT.

We also described the results of pre-existed significant CAS reviewed from the CT/MR images before RT in the RESULTS section.

Results, page 9, line 20-22

The mean interval from the RT completion to the last follow up were similar between the two groups. None of the patients noted with significant CAS during follow-up had pre-existed lesion prior to RT.

However, due to the source images and axial cuts of the head and neck CT/MRI may not be as good as the CTA/MRA, we described this in the “limitation paragraph”.

DISCUSSION, page 18, line 19-23

Third, the contrast enhanced CT or MRI studies were reviewed to screen for secondary outcomes and pre-existed vascular lesions prior to RT. The timing of contrast injection was less suitable for vascular evaluation, which could have underpowered the results of vascular evaluation and could be a source of bias. A prospective study may help to validate our conclusions in the future.

  1. (2) the lack of data on doses to carotid arteries.

Answer: Thank you for your comments, that is indeed a critical weakness particularly when the aim of the study was to discuss the casual relationship between RT method and CAS development, or the underlying mechanism of RT related CAS.

However, the major objective of this study was to propose a useful and more precise vascular follow-up strategy in these HNC patients after RT. More importantly, we hoped to alert the clinician to beware of the higher risks of significant CAS in the non-NPC patients, even in early years after RT. Non-NPC and NPC groups may represent the patients received different treatment method, associated vascular risk factors, radiation doses to carotid arteries, and socioeconomic status. Types of HNC would be a practical clinical marker to select the vulnerable patients, since doses to carotid arteries are not easily acquired for most of the primary clinician, especially for non-radio-oncologists.

We revised our manuscript to address this limitation.

Discussion section/ limitations.

Page 15, line 21-23

Different types of HNC in this study may represent patients the patient population with various treatment methods, vascular risk factors, co-morbidities, diverse life styles, and different socio-economic composition.

Page 16, line 24 – page 17, line 4

Moreover, the radiation doses to carotid arteries may have impacts on the CAS development [2, 34]. Without the dosimetric data of carotid arteries, our results were insufficient to discuss whether the difference of CAS risks between the two groups was associated with diverse radiation doses to carotid arteries. In general. the associated outcomes in this study could be related to the combination of multiple factors. Further studies maybe needed to discuss underlying mechanisms.

Page 18, line 4-8

In the present study, method of RT (PBT or VMAT) was not associated with a higher risk of significant CAS in our data (AHR = 1.30, 95% CI: 0.39-4.38, p = 0.67; supplementary table), however our results could be limited due to the lack of comprehensive dosimetric data in the study. Further clinical trials are warranted to discuss this issue. 

Page 18, line 23 – page 18, line 4

Fourth, we reviewed the data regarding details of RT retrospectively and therefore the dosimetric data to carotid arteries may not be available. This may have vital impact on discussing the casual relationship between CAS and types of HNC. However, the aim of this research was not to discuss the pathogenesis why non-HNC patients are vulnerable to CAS development but was to provide a more precise clinical follow-up strategy of vascular complication between different HNCs.

Due to the retrospective nature, we were not able to calculate most of the patients’ dosmetric data. However, we tried to calculate available the doses to common carotid artery in some example cases.

We noticed the radiation doses to the carotid arteries were variable and unable to give a conclusive answer. Under your kind remind, we would like recruit these data in our ongoing prospective registry. We hope this may help to discuss the casual relationship of RT and post RT complications more detailed in the future. We hope to express our appreciation again.

  1. The hypothesis that non-NPC patients are more prone to CAS after RT therefore cannot be tested without having pre-RT CDU data. Since it was shown also by the authors that non-NPC patients have significantly more non-RT related risk factors for developing CAS compared to NPC patients, the conclusion of the paper that non-NPC patients are more prone to CAS after RT is misleading. These patients are more prone to CAS even before RT and even if they would be treated without RT. These issues will be discussed below along with the others.

Answer: Thank you for your kindly remind. We also agree that casual-relationship should be interpretated cautiously in a retrospective cohort study, particularly when the pre-RT CDU data was not available. Due to the data were acquired from the routine practice, and pre-RT vascular screen was not an evidence-support practice in previous guideline. We are sorry that we do not have pre0RT CDU data. To minimize this bias, we described how we excluded pre-existing carotid stenosis in the revised manuscript as previous mentioned. We also revised the conclusion to avoid over-interpretation of the casual relationship between cancer types and carotid stenosis.

Abstract

Methods: We enrolled 496 patients with HNC who had received their final RT dose in our hospital. These patients underwent carotid duplex ultrasound (CDU) for monitoring significant carotid artery stenosis (CAS). Brain imaging were reviewed to detect vertebral, intracranial artery stenosis, or preexisted CAS before RT. Primary outcome was significant CAS at the internal or common carotid artery within the first 5 years after RT. We categorized the patients into nasopharyngeal carcinoma (NPC) and non-NPC groups and compared the cumulative occurrence of significant CAS between the groups using Kaplan–Meier and Cox-regression analyses.

Results: Compared to the NPC group, the non-NPC group had a higher frequency of significant CAS (12.7% vs. 2.0%) and were more commonly associated with significant CAS after adjusting the covariates (Adjusted hazard ratio: 0.17, 95% confident interval: 0.05-0.57) during the follow-up period. All the non-NPC subtypes (oral cancer/oropharyngeal, hypopharyngeal, and laryngeal cancers) were associated with higher risks of significant CAS than the NPC group (p < 0.001 respectively).

Conclusion: Significant CAS was more frequently noted within 5 years of RT among the patients with non-NPC HNC than among the patients with NPC.

Discussion

Our study revealed that patients with non-NPC HNC were associated with a higher risk of significant CAS diagnosis within 5 years after RT completion.

Conclusion, page 19, line 14

Significant CAS within 5 years of RT completion was noted more frequently among the patients with non-NPC HNC than among those with NPC.

  1. Specific comments (in order of appearance regardless of their importance)

*the lines of the manuscript are not enumerated

**the abstract should be corrected according to the corrections in the main text and will not be commented on separately

Answer: thank you, we’ve revised the abstract according to the correction in the main text.

In further, we also revised the GRAPH-ABSTRACT

  1. Introduction:

- “Head and neck cancer (HNC) is a group of cancers developing in the soft tissues, salivary glands, or the mucosa of the upper respiratory or digestive system that cover the oral and nasal cavities.” o The term Head and neck cancer includes also cancers of the laryngeal, pharyngeal and paranasal sinuses' mucosa. This statement needs to be corrected.

Answer: Thank you for your kind remind, we’ve corrected the statement accordingly.

Introduction, page 3, line 2-5

Head and neck cancer (HNC) is a group of cancers developing in the soft tissues, salivary glands, the mucosa of the upper respiratory, or digestive system that cover the oral and nasal cavities, laryngeal, pharyngeal, hypopharyngeal, and paranasal sinuses' mucosa.

  1. Therefore, PBT may reduce the integral radiation dose to normal tissues, thereby avoiding collateral damage. However, the incidence of cervical–cranial vascular complications of these advanced RT methods remain unknown. In the present study, we investigated the frequency of cervical–cranial vascular complications under advanced RT methods among patients with various HNC types in current era.” o Without assessing the doses to carotid arteries the rationale of this study is questionable.

Ans: Thank you. We agree with your opinion. Our previous version of manuscript was confusing and did not reflect the true aim of our study. We revised the manuscript accordingly.

Introduction, page 4, line 7-8

However, the incidence of cervical–cranial vascular complications in the patients undergoing these advanced RT methods remain unknown. In the present study, we investigated the frequency of cervical–cranial vascular complications among patients with various HNC types in current era and hoped to develop a precise clinical follow-up strategy of vascular complication between different HNC types.

  1. Methods:

- It should be explicitly stated that this was a retrospective study.

Answer: thank you for your comment. We stated that this study was a retrospective study in the revised manuscript.

Methods, page 4, line 17-18

Patient recruitment and demographic data

Our cohort included patients with HNC who had ever undergone RT between November 1, 2011, and October 31, 2021, and who received regular follow-ups at the radio-oncology and neurology departments of Chang Gung Memorial Hospital at Linkou. Because we aimed to investigate neurovascular complications during the early phase after RT and hope to minimize the bias caused by the variance of RT methods in different era. We retrospectively reviewed the patients completed their RT after January 01, 2015 in this study,

  1. - Were only patients receiving definitive RT with radical intent included or were there also patients receiving postoperative RT? This has major implications as neck surgery can be detrimental to carotid arteries.

- Were the patient that received induction chemotherapy (ChT) also included? How many patients received concomitant ChT and what ChT scheme was used? Vasculopathy of chemotherapeutics is not to be neglected.

Answer: Thank you for your comment. We described the treatment protocols of different HNC types in the Methods section. We mentioned the influence of surgery to the carotid artery disease in the Discussion Section. Regarding the influence of chemotherapy on vasculopathy, since the most of the patients received the similar platinum-based CCRT, the influence could be minor.

We added a paragraph regarding the treatment protocol of different HNC cancer in the method session.

Methods, page 5, line 16-page 6, line 5

Cancer treatment and RT data

The standard treatment in our hospital was the definitive RT for the NPC patients, while around four in five of the larynx, hypopharynx, and oropharynx cancer patients received definitive RT for initial organ preservation as well. Differently, most of the oral cavity cancer patients received postoperative RT in our hospital, and definitive RT was the alternative treatment for patients who were not suitable for surgical treatment. For all the HNC patients receiving RT, platinum-based (Cisplatin or carboplatin) concurrent chemotherapy was the main regimen.

We also discuss the influence of these factors to the results in the Discussion Section.

In further, treatment methods could be different among these HNC types. Oral cavity cancer patients mainly receive surgical treatment followed by post-operative RT while NPC patients majorly receive definitive RT. It is known that RT along has higher risks of CAS than surgical treatment in HNC patients [4]. Surgical treatment in addition to RT could possibly injures carotid arteries [31]. Chemotherapy regimens may also lead to vascular toxicity [32]. However, this may have minor influence to our results due to most of the HNC patients received platinum-based concurrent chemoradiation therapy in our hospital. Moreover, the radiation doses to carotid arteries may also have impacts on the CAS development [2, 33]. Without the dosimetric data of carotid arteries, our results were insufficient to discuss whether the difference of CAS risks between the two groups was associated with diverse radiation doses to carotid arteries. Therefore. the associated outcomes in this study could be related to the combination of multiple factors. Further studies maybe needed to discuss underlying mechanisms.

  1. We also reviewed the patients’ pretreatment magnetic resonance imaging (MRI) data to exclude patients who had presented with significant CAS or vertebral artery (VA) stenosis before RT.” o It stated by the authors below that significant CAS was defined as ">50% stenosis on B-mode images with a peak systolic velocity ≥120 cm/s based on the hemodynamic criteria for any internal or common carotid artery in the carotid duplex ultrasound (CDU) examination". Therefore CAS could not be excluded before RT using only MRI according to the description of the methodology in this manuscript. Alternatively, authors could add the description of diagnosing CAS on MRI. This is an important drawback of this study. Furthermore, this exclusion criteria, significant CAS, is not presented in Figure 1. Therefore no patients had CAS before RT and so none were excluded for this reason?

Answer: thank you for your comment, as we described in previous revision letter, we revised the METHOD/RESULTS sections according to your suggestion.

Methods, page 7, line 3-10

Routine CDU evaluation prior to the initiation of RT was not recommended in previous guideline [19], we reviewed the images of carotid and vertebral arteries (VA) in the contrast enhanced CT and/or magnetic resonance imaging (MRI) performed before the initiation of cancer treatment to detect preexisted significant CAS or VA stenosis. We defined as a pre-existed significant CAS/ VA stenosis if there were any plaques/stenosis occupied more than 50% of the vessel lumen in the axial views. We hoped this could help to verify whether the significant CAS developed after RT or existed prior to RT.

There was no patient had pre-existed significant CAS, we also described the results of pre-existed significant CAS reviewed from the CT/MR images before RT in the RESULTS section.

Results, page 9, line 20-22

The mean interval from the RT completion to the last follow up were similar between the two groups. None of the patients noted with significant CAS during follow-up had pre-existed lesion prior to RT.

Due to the source images and axial cuts of the head and neck CT/MRI may not be as good as the CTA/MRA, we described this in the “limitation paragraph”.

DISCUSSION, page 18, line 19-23

Third, the contrast enhanced CT or MRI studies were reviewed to screen for secondary outcomes and pre-existed vascular lesions prior to RT. The timing of contrast injection was less suitable for vascular evaluation, which could have underpowered the results of vascular evaluation and could be a source of bias. A prospective study may help to validate our conclusions in the future.

  1. Follow-ups: “...CBC...” → needs explanation

Answer: thank you, we added the explanation.

Methods, page 7, line 19-21

Upon the first visit at neurology department, the patients’ laboratory data including complete blood count, blood chemistry, lipid profile and thyroid hormone levels were checked.

  1. “MRI was also used to determine each patient’s vascular status.” See above. What is the significance of these MRI for vascular status? Was MRI used to verify diagnosis of CAS in all the patients? How many patients had CDU and how many MRI for CAS assessment? How many had CAS diagnosed with CDU that were later found to have no CAS on MRI and vice versa?

Answer: Thank you for your comment. We are indeed sorry that our previous manuscript made the readers confused. We revised the method session accordingly (as mentioned before).

Methods, page 7, line 4- page 8, line 5

Routine CDU evaluation prior to the initiation of RT was not recommended in previous guideline [19], we reviewed the images of carotid and vertebral arteries (VA) in the contrast enhanced CT and/or magnetic resonance imaging (MRI) performed before the initiation of cancer treatment to detect preexisted significant CAS or VA stenosis. We defined as a pre-existed significant CAS/ VA stenosis if there were any plaques/stenosis occupied more than 50% of the vessel lumen in the axial views. We hoped this could help to verify whether the significant CAS developed after RT or existed prior to RT.

The secondary outcomes were the presence of significant vertebral artery (VA) stenosis, significant intracranial arterial stenosis (ICAS), and occurrence of symptomatic ischemic stroke (IS). Types of carotid interventions were also recorded.

The patients received regular follow-up at the radio-oncology department at least every 6 months after RT. Contrast enhanced CT and/or MRI were arranged in the meantime to identify potential distal metastasis or recurrence of the primary tumor. To monitor the neurovascular complications, patients were also referred to our neurology out-patient department during the post-RT period. Upon the first visit at neurology department, the patients’ laboratory data including complete blood count, blood chemistry, lipid profile and thyroid hormone levels were checked. Meanwhile, patients underwent the first carotid duplex ultrasound (CDU) study to screen the presence of significant CAS. Total plaque scores (TPS) [6], intimal medial thickness (IMT) of the common carotid arteries, and degree of CAS were also measured in this examination. Patients received repeated CDU exam 12-24 months later based on the severity of carotid artery lesions in the first visit [19]. We also reviewed the patients’ CT/MR images arranged for cancer follow-ups to find the possible signs of secondary outcomes.

In patients noted with a sign of significant CAS on any CDU study, significant VA stenosis or ICAS, we then arranged CT angiography or MR angiography to confirm the diagnosis.

  1. “The primary outcome in this study was the diagnosis of significant CAS.” o At what time point? And how often? This is of outmost importance.

Answer:thank you for your comment, we corrected the definition of primary outcome.

Abstract:

Primary outcome was significant CAS at the internal or common carotid artery within the first 5 years after RT.

Method, page 6, line 24-25

Outcomes and follow-ups

The primary outcome in this study was the diagnosis of significant CAS within the first 5 years after RT.

  1. ...conducted a Kolmogorov–Smirnov test to evaluate normality.” o The results of this test should be presented in the Results section.

Answer: Thank you for your comment. Sorry for the confusion, because the sample size of this study was large (close to 500), most of the formal normality tests (such as the Kolmogorov–Smirnov test) may not be passed, so we decided to remove this sentence.

Methods, page 8, line 7-9

We used SPSS 22.0 (SPSS, Chicago, IL, USA) to analyze the clinical data. Parameters are presented as means ± standard deviations or frequencies (%).

  1. “We also used Cox regression analysis to...” o The authors should provide more details about the statistical analyses as per TRIPOD3, specifically: â–ª Describe how predictors were handled in the analyses. â–ª Specify type of model and model-building procedures (including any predictor selection)  

Ans: Thank you. We revised the statistical analysis session accordingly. We hoped the revised version may better show how we handle the predictors in the analyses and provide more details about the statistical analyses as per TRIPOD3.

Methods, page 8, line 7-21

Statistical analysis

   We used SPSS 22.0 (SPSS, Chicago, IL, USA) to analyze the clinical data. Parameters are presented as means ± standard deviations or frequencies (%). We used an independent two-sample t test to identify differences in the continuous variables between the study groups. The categorical variables were compared using a chi-square test or Fisher’s exact test. Event risk and time to significant CAS were compared between the study groups through Kaplan–Meier analysis with log-rank test. According to previous reports, we selected known predictors factors for atherosclerosis-associated CAS (smoking, diabetes mellitus, dyslipidemia, hypertension, coronary artery disease, and glycated hemoglobin) and RT-associated CAS (RT doses, types of RT, and re-irradiation) in the Cox regression model [6, 20]. We analyzed continuous variables (RT dose and glycated hemoglobin) as continuous data instead of categorized them into groups. We used univariate Cox regression model and multivariable Cox regression model with backward selection to see the relationship between these 9 risk factors, NPC, and CAS risk. Significance was indicated by p < 0.05.

  1. If done, report the unadjusted association (univariate analysis) between each candidate predictor and outcome.

Answer: Thank you for your comment, we provided these data in the supplementary table 2.

  1. Importantly, all the covariates that were included in the multivariate model should be listed in the Results section where reporting on the results of the multivariate analyses.

Answer: Thank you, we listed these results into the Result section.

Results, page 10, line 3-6

Of notes, the glycated hemoglobin level (AHR = 1.03, 95% CI 1.01-1.06, p = 0.03) and history of re-irradiation (Reirradiation vs. non-reirradiation: 9.7% vs. 8.1%, AHR = 5.95, 95% CI 1.25-28.29, p = 0.01) were both associated with higher risks of significant CAS in the Cox regression model (Supplementary table).

  1. Even though the “Rule of Ten Events per Variable” is known to probably be too strict, is seems that the number of events per variable (EPV) in the CAS analysis in this manuscript was very low, 41 in total.4 I kindly ask the authors to address this issue in the response to this review. This does not necessarily have to be included in the manuscript.

Answer: Thank you for your kind remind.

According to our final multivariate results, 3 significant predictors were retained at the end, so according to the "Rule of Ten Events per Variable", we have met this criterion.

  1. Dyslipidemia was included in the multivariate analysis along with lipoproteins. â–ª

How was dyslipidemia defined?

Answer: thank you, we added the definition of dyslipidemia in the method session.

Methods, page 5, line 1-3

Dyslipidemia was defined when patients’ low-density lipoprotein cholesterol (LDL) level was ≧130 mg/dl or when they had received lipid lowering therapy at first visit in neurology department.

  1. Measuring dyslipidemia and lipoproteins is a duplication and leads to multicollinearity problem in multivariable models.

Answer: thank you for your comment., we finally removed lipo-protein in multivariable models because of collinearity and we updated the statistic method.

Methods, page 8, line 13-17

According to previous reports, we selected known predictors factors for atherosclerosis-associated CAS (smoking, diabetes mellitus, dyslipidemia, hypertension, coronary artery disease, and glycated hemoglobin) and RT-associated CAS (RT doses, types of RT, and re-irradiation) in the Cox regression model [6, 20].

  1. Smoking o How was this defined? Current vs. all others? Current and former vs. never-smokers? Other?

Answer: Thank you for your suggestion, we’ve revised the manuscript accordingly.

Methods, page 5, line 3-6

We defined the patients as having histories of cigarette smoking and betel quid chewing if they were current or former users.

  1. Results

 “Finally, 496 HNC patients had undergone at least one carotid vascular study...” o Specify at which time the CDU was performed, e.g. range and median.

Answer: thank you, we add the mean time of the first CDU study of the two groups.

Results, page 9, line 16-17, and table 1

The mean time interval between RT completion and the first CDU were similar between the two groups (21.79±14.14 vs. 22.82±14.35 months, p =0.44).

  1. Doses to carotids are vital. Without this data the manuscript is of dubious value.

Answer: Thank you for your suggestion. We agree with your opinion. We addressed this limitation in the discussion session.

Discussion section/ limitations.

Page 15, line 21-23

Different types of HNC in this study may represent patients the patient population with various treatment methods, vascular risk factors, co-morbidities, diverse life styles, and different socio-economic composition.

Page 16, line 24 – page 17, line 4

Moreover, the radiation doses to carotid arteries may have impacts on the CAS development [2, 34]. Without the dosimetric data of carotid arteries, our results were insufficient to discuss whether the difference of CAS risks between the two groups was associated with diverse radiation doses to carotid arteries. In general. the associated outcomes in this study could be related to the combination of multiple factors. Further studies maybe needed to discuss underlying mechanisms.

Page 18, line 4-8

In the present study, method of RT (PBT or VMAT) was not associated with a higher risk of significant CAS in our data (AHR = 1.30, 95% CI: 0.39-4.38, p = 0.67; supplementary table), however our results could be limited due to the lack of comprehensive dosimetric data in the study. Further clinical trials are warranted to discuss this issue. 

Page 18, line 23 – page 18, line 4

Fourth, we reviewed the data regarding details of RT retrospectively and therefore the dosimetric data to carotid arteries may not be available. This may have vital impact on discussing the casual relationship between CAS and types of HNC. However, the aim of this research was not to discuss the pathogenesis why non-HNC patients are vulnerable to CAS development but was to provide a more precise clinical follow-up strategy of vascular complication between different HNCs.

  1. - “the mean total radiation doses were similar between the two groups (Table 1)” o In the previous paragraph the authors stated the doses were different. Needs revision.

Answer: thank you, we corrected the errors.

  1. Table 1

- It should be provided in the table (or below) which statistical test was used.

Answer: thank you, we’ve added which statistical test was used in table 1.

Table 1.

CDU, carotid duplex ultrasound; HDL, high-density lipoprotein; IMT, intimal medial thickness; LDL, low-density lipoprotein; NPC, nasopharyngeal carcinoma; RT, radiation therapy; p < 0.05.

Data were examined by two-sample t-tests (continuous variables) and chi-square tests (categorical variables).

  1. Cancer types o This makes no sense. The authors compared cancer types, NPC and non-NPC, between NPC and non-NPC groups (p<0.01!!!). This makes no sense.

Answer: Thank you for the suggestion.

We pleased you to let us keep the proportion of cancer types, because this helps the reader to know how many oral cavity cancer/ laryngeal/hypopharyngeal cancers were there in this data.

But we deleted the p < 0.01 based on your comment. We agree that this could be non-sense.

  • -26. Free T4 → units are missing

Answer: We’ve corrected this error.

  1. Figure 3

- These curves are not Cox regression. The calculated HR could be the result of Cox, but the curves are not. Annotations needs to be corrected. More importantly, how does this Figure 3 differ from Figure 2 and what important data does it add to Figure 2 that it cannot be simply added to Figure 2 or to the text?

Ans: thank you for your comment. We agree with your opinion and deleted the figure 3. In the revised version, the new figure 3 is the figure 4 in the previous version.

  1. Table 2

- It should be provided in the table (or below) which statistical test was used.

Answer: thank you, we’ve added the statistical tests.

  1. Supplementary table

- Hazard ratios of what? This table needs a description.

Answer: thank you, revised the title of supplementary table.

New title: Associated factors of significant CAS in Cox regression models

  1. Why were the first three variables chosen for multivariate analysis? See above the TRIPOD comment.

Answer: thank you for your suggestion, we revised our manuscript to follow the TRIPOD comment.

Method, page 8, line 13-19

According to previous reports, we selected known predictors factors for atherosclerosis-associated CAS (smoking, diabetes mellitus, dyslipidemia, hypertension, coronary artery disease, and glycated hemoglobin) and RT-associated CAS (RT doses, types of RT, and re-irradiation) in the Cox regression model [6, 20]. We analyzed continuous variables (RT dose and glycated hemoglobin) as continuous data instead of categorized them into groups.

  1. "Multi-variates" by itself provides little information. The authors should consider "multivariate analysis" or similar for the third column.

Answer: Thank you for your suggestion, we’ve revised the third column according to your suggestion.

Round 2

Reviewer 1 Report

Thank you very much for the elaborate reply to the review. Most issues have been addressed properly. One issue remains that should be addressed  by the authors: The following sentence was just copied to the corrected version of the manuscript: "The paper would gain substantially for instance by providing a proposal for a prospective study, especially by including proposals on stratification factors."

My intention was to encourage the authors to provide a "theoretical" study design and set-up for a possible future study. As the awareness of radiation oncologists regarding the topic of CAS can be improved by the manuscript it would be valuable to present a study proposal and adress the pitfalls in order to allow for properly designed future trials.

Author Response

Dear Reviewer

Thank you for your detailed review. Your thorough and excellent suggestions not only made our manuscript better but also guided us to improve the quality of our ongoing cohort data.

 We have provided a point-by-point revision and our responses to all your comments. The reasons and revisions are provided below. In the revised manuscript, all the changes are highlighted. We deeply appreciate your valuable review, which stimulated a more thorough consideration of the article. Thank you very much and we hope the revised manuscript became much better and reached the standard of JPM.

Sincerely Yours,

Yeu-Jhy Chang, MD

Stroke Center and Department of Neurology

Chang Gung Memorial Hospital, Linkou Medical Center and College of Medicine, Chang Gung University, Taoyuan, Taiwan

No. 5, Fu-Hsing ST. Kueishan, Taoyuan, 33333 Taiwan

Tel: 886-3-3281200 ext 8340

And 

Joseph Tung-Chieh Chang MD, MHA

Department of Radiation Oncology, Proton and Radiation Therapy Center, Chang Gung Medical Foundation, Linkou Chang

Gung Memorial Hospital, Taoyuan, Taiwan.

  1. Thank you very much for the elaborate reply to the review. Most issues have been addressed properly.

Answer: thank you, we are so grateful to have the opportunity to have your review. Your thorough and excellent suggestions not only made our manuscript better but also guided us to improve the quality of our ongoing cohort data.

  1. One issue remains that should be addressed by the authors: The following sentence was just copied to the corrected version of the manuscript: "The paper would gain substantially for instance by providing a proposal for a prospective study, especially by including proposals on stratification factors."

My intention was to encourage the authors to provide a "theoretical" study design and set-up for a possible future study. As the awareness of radiation oncologists regarding the topic of CAS can be improved by the manuscript it would be valuable to present a study proposal and adress the pitfalls in order to allow for properly designed future trials.

Answer: thank you very much. We are sorry for our misunderstanding. We’ve deleted this sentence.

Fifth, there were a wide range of confounders regarding the presented data. Our conclusions might be at high risk of selection and reporting bias due to the large number of influencing factors and missing data as well as the large inhomogeneity of the patient cohort. The major factor of inhomogeneity is the above-mentioned issue of imaging disparities. Finally, the generalizability of our results to patients of other ethnicities remains to explore. Due to the above limitations, these results were only hypothesis generating.

Reviewer 2 Report

I thank the authors for their extensive reply. The authors have more than sufficiently addressed all the issues. Importantly, they further underlined the limitations of the study, which only adds to its credibility.

Minor suggestions:

- Page 18: "However, the aim of this research was not to discuss the pathogenesis why non-HNC non-NPC patients are vulnerable to CAS development but was to provide a more precise clinical follow-up strategy of vascular complication between different HNCs."

- English needs to be revised, e.g. p.5: "Surgical treatment in addition to RT could possibly injures carotid arteries", and many similar small mistakes, however, the manuscript is nonetheless understandable.

- Kind suggestion for the future: in your next paper, which I am looking forward to, please adhere to TRIPOD guidelines on reporting the MVA (https://bmcmedicine.biomedcentral.com/articles/10.1186/s12916-014-0241-z)

Author Response

# Reviewer 2

Dear Reviewer

Thank you for your detailed review. We have provided a point-by-point revision and our responses to all your comments. The reasons and revisions are provided below. In the revised manuscript, all the changes are highlighted. We deeply appreciate your valuable review, which stimulated a more thorough consideration of the article. Thank you very much and we hope the revised manuscript became much better and reached the standard of JPM.

Sincerely Yours,

Yeu-Jhy Chang, MD

Stroke Center and Department of Neurology

Chang Gung Memorial Hospital, Linkou Medical Center and College of Medicine, Chang Gung University, Taoyuan, Taiwan

No. 5, Fu-Hsing ST. Kueishan, Taoyuan, 33333 Taiwan

Tel: 886-3-3281200 ext 8340

And 

Joseph Tung-Chieh Chang MD, MHA

Department of Radiation Oncology, Proton and Radiation Therapy Center, Chang Gung Medical Foundation, Linkou Chang

Gung Memorial Hospital, Taoyuan, Taiwan.

  1. I thank the authors for their extensive reply. The authors have more than sufficiently addressed all the issues. Importantly, they further underlined the limitations of the study, which only adds to its credibility.

Answer: thank you, we are so grateful to have the opportunity to have your review. Your thorough and excellent suggestions not only made our manuscript better but also guided us to improve the quality of our ongoing cohort data.

Minor suggestions:

  1. - Page 18: "However, the aim of this research was not to discuss the pathogenesis why non-HNC non-NPC patients are vulnerable to CAS development but was to provide a more precise clinical follow-up strategy of vascular complication between different HNCs."

Answer: thank you for your correction. We’ve corrected this manuscript accordingly.

However, the aim of this research was not to discuss the pathogenesis why non-NPC HNC patients are vulnerable to CAS development but was to provide a more precise clinical follow-up strategy of vascular complication between different HNCs.

  1. - English needs to be revised, e.g. p.5: "Surgical treatment in addition to RT could possibly injures carotid arteries", and many similar small mistakes, however, the manuscript is nonetheless understandable.

Answer: thank you for your remind, we’ve corrected the grammar error.

Surgical treatment in addition to RT could possibly injure carotid arteries

We also corrected other grammar, or spelling errors in the manuscript

Such as

Abstract

To investigate the frequency of cervical–cranial vascular complications soon after radiation therapy (RT) and identify differences

Primary outcome was significant CAS at the internal or common carotid artery within first 5 years after RT

Introduction

RT-induced CAS is typically widespread [6], rapidly progressive [7], and usually affects the common carotid artery (CCA)

Methods

Outcomes and follow-ups

The primary outcome in this study was the diagnosis of significant CAS within first 5 years after RT.

Upon the first visit at neurology department, the patients’ laboratory data including complete blood count, blood chemistry tests,

We also reviewed the patients’ CT/MR images arranged for cancer follow-ups to find the possible clues of secondary outcomes.

Results

Among the 496 patients, 201 (40.5%) and 295 (59.5%) were categorized into the NPC and non-NPC groups (Figure 1).

All the patients with significant CAS did not have pre-existed significant CAS before RT,

Discussion

The non-NPC HNC patients had lower level of LDL at enrolment. It is possible that the non-NPC HNC,

In the present study, methods of RT (PBT or VMAT) were not associated with a higher risk of significant CAS in our data

  1. - Kind suggestion for the future: in your next paper, which I am looking forward to, please adhere to TRIPOD guidelines on reporting the MVA (https://bmcmedicine.biomedcentral.com/articles/10.1186/s12916-014-0241-z)

Answer: thank you for your suggestion, we’ll advance our research quality in the future. We hope our ongoing research can help to draw the clinician’s attention to this issue.